# DUE: End-to-End Document Understanding Benchmark

**Łukasz Borchmann**[*,ℵ,†]    **Michał Pietruszka**[*,ℵ,‡]    **Tomasz Stanisławek**[*,ℵ,§]

**Dawid Jurkiewicz**[ℵ,¶]    **Michał Turski**[ℵ,¶]    **Karolina Szyndler**[ℵ]    **Filip Graliński**[ℵ,¶]

[ℵ]Applica.ai
`firstname.lastname@applica.ai`

[†]Poznan University of Technology          [‡]Jagiellonian University, Krakow

[§]Warsaw University of Technology          [¶]Adam Mickiewicz University, Poznan

## Abstract

Understanding documents with rich layouts plays a vital role in digitization and hyper-automation but remains a challenging topic in the NLP research community. Additionally, the lack of a commonly accepted benchmark made it difficult to quantify progress in the domain. To empower research in this field, we introduce the Document Understanding Evaluation (DUE) benchmark consisting of both available and reformulated datasets to measure the end-to-end capabilities of systems in real-world scenarios. The benchmark includes Visual Question Answering, Key Information Extraction, and Machine Reading Comprehension tasks over various document domains and layouts featuring tables, graphs, lists, and infographics. In addition, the current study reports systematic baselines and analyzes challenges in currently available datasets using recent advances in layout-aware language modeling. We open both the benchmarks and reference implementations and make them available at `https://duebenchmark.com` and `https://github.com/due-benchmark`.

## 1   Introduction

While mainstream Natural Language Processing focuses on plain text documents, the content one encounters when reading, e.g., scientific articles, company announcements, or even personal notes, is seldom plain and purely sequential. In particular, the document's visual and layout aspects that guide our reading process and carry non-textual information appear to be an essential aspect that requires comprehension. These layout aspects, as we understand them, are prevalent in tasks that can be much better solved when given not only text sequence on the input but pieces of multimodal information covering aspects such as text-positioning (i.e. location of words on the 2D plane), text-formatting (e.g., different font sizes, colors), and graphical elements (e.g., lines, bars, presence of figures) among others. Over the decades, systems dealing with document understanding developed an inherent aspect of multi-modality that nowadays revolves around the problems of integrating visual information with spatial relationships and text [36, 2, 50, 13].

---

[*]Equal contribution

35th Conference on Neural Information Processing Systems (NeurIPS 2021) Track on Datasets and Benchmarks.

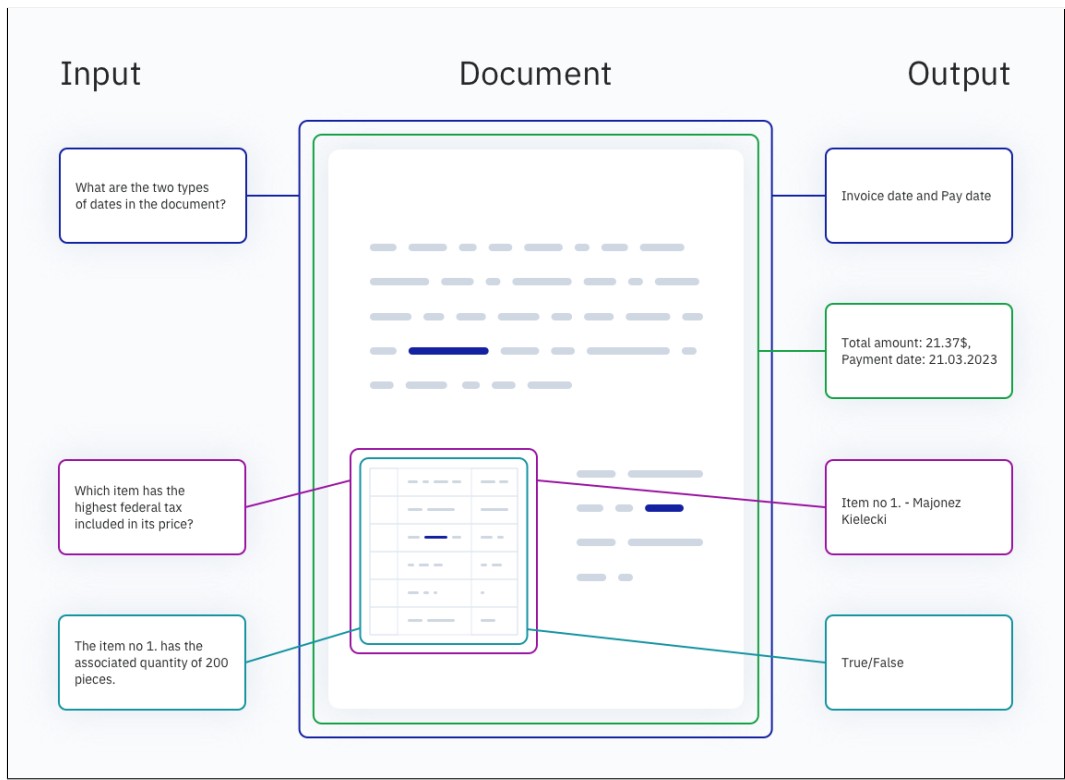

Figure 1: Document Understanding covers problems ranging from the ■ extraction of key information, through ■ verification statements related to rich content, to ■ ■ answering open questions regarding an entire file. It may involve the comprehension of multi-modal information conveyed by a document.

In general, when document processing systems are considered, the term *understanding* is thought of specifically as the capacity to convert a document into meaningful information [10, 57, 16]. This fits into the rapidly growing market of hyperautomation-enabling technologies, estimated to reach nearly $600 billion in 2022, up 24% from 2020 [42]. Considering that unstructured data is orders of magnitude more abundant than structured data, the lack of tools necessary to analyze unstructured data and extract structured information can limit the performance of these intelligent services. The process of structuring data and content must be robust to various document domains and tasks.

Despite its importance for digital transformation, the problem of measuring how well available models obtain information from a wide range of tasks and document types and how suitable they are for freeing workers from paperwork through process automation is not yet addressed. Meanwhile, in other research communities, there are well-established progress measuring methods, like the most recognizable NLP benchmarks of GLUE and SuperGLUE covering a wide range of problems related to plain-text language understanding [53, 52] or VTAB and ImageNet in the computer vision domain [59, 11]. We intend to bridge this major gap by introducing the first Document Understanding benchmark (available at `https://duebenchmark.com`).

It includes tasks that either originally had a vital layout understanding component or were reformulated in such a way that after our modification, they require layout understanding. In particular, there is no structured representation of the underlying text, such as a database-like table given in advance, and it has to be determined from the input file as a part of the end-to-end process. Every time, there is only a PDF file provided as an input. Additionally, for the convenience of other researchers, we provide information about textual tokens and their locations (bounding boxes) which are coming from the OCR system or directly from the born-digital PDF file (see Section 4).

**Contribution.** The idea of the paper is to gather, reformulate and unify a set of intuitively dissimilar tasks that we found to share the same underlying requirement of understanding layout concepts. In order to organize them in a useful benchmark, we contributed by performing the following steps:

1. We reviewed and selected the available datasets. Additionally, we reformulated three tasks to a document understanding setting and obtained original documents for all of them (PWC, WTQ, TabFact).

2. We performed data cleaning, including the improvements of data splits (DeepForm, WTQ), data deduplication, manual annotation (PWC, DeepForm), and converted data to a unified format (all datasets).

3. We implemented competitive baselines and measured human performance where it was required (PWC, DeepForm, WTQ).

4. We identified challenges related to the current progress in the DU domain's tasks and provided manually annotated diagnostic sets (all datasets).

These contributions are organized and described in Table 2. Additionally, a wider review of available tasks is described in Appendix A.

## 2 The state of Document Understanding

We treat Document Understanding as an umbrella term covering problems of Key Information Extraction, Classification, Document Layout Analysis, Question Answering, and Machine Reading Comprehension whenever they involve rich documents in contrast to plain texts or image-text pairs (Figure 1).

In addition to the problems strictly classified as Document Understanding, several related tasks can be reformulated as such. These provide either text-figure pairs instead of real-world documents or parsed tables given in their structured form. Since both can be rendered as synthetic documents with some loss of information involved, they are worth considering bearing in mind the low availability of proper Document Understanding tasks.

### 2.1 Landscape of Document Understanding tasks

**KIE.** Key Information Extraction, also referred to as Property Extraction, is a task where tuple values of the form (property, document) are to be provided. Contrary to QA problems, there is no question in natural language but rather a phrase or keyword, such as *total amount*, or *place of birth*. Public datasets in the field include extraction performed on receipts [20, 38], invoices, reports [45], and forms [24]. Documents within each of the mentioned tasks are homogeneous, whereas the set of properties to extract is limited and known in advance – in particular, the same type-specific property names appear in both test and train sets. In contrast to Name Entity Recognition, KIE typically does not assume that token-level annotations are available, and may require normalization of values found within the document.

**Classification.** Classification in our context involves rich content, where comprehension of both visual and textual aspects is required since unimodal models underperform. Though document image classification was initially approached using solely the methods of Computer Vision, it has recently become evident that multi-modal models can achieve significantly higher accuracy [55, 56, 40]. Similar conclusions were recently reached in other tasks, e.g., assigning labels to excerpts from biomedical papers [54].

**DLA.** Document Layout Analysis, performed to determine a document's components, was initially motivated by the need to optimize storage and the transmission of large information volumes [36]. Even though its motivation has changed over the years, it is rarely an end in itself but rather a means to achieve a different goal, such as improving OCR systems. A typical dataset in the field assumes detection and classification of page regions or tokens [61, 30].

**QA and MRC.** At first glance, Question Answering and Machine Reading Comprehension over Documents is simply the KIE scenario where a question in natural language replaced a property name. More differences become evident when one notices that QA and MRC involve an open set of questions and various document types. Consequently, there is pressure to interpret the question and

to possess better generalization abilities. Furthermore, a specific content to analyze demands a much stronger comprehension of visual aspects, as the questions commonly relate to figures and graphics accompanying the formatted text [33, 32, 49].

**QA over figures.** Question Answering over Figures is, to some extent, comparable with QA and MRC over documents described above. The difference is that a 'document' here consists of a single born-digital plot, reflecting information from chosen, desirably real-world data. Since questions in this category are typically templated and figures are synthetically generated by authors of the task, datasets in this category contain as many as millions of examples [34, 4].

**QA and NLI over tables.** Question Answering and Natural Language Inference over Tables are similar, though in the case of NLI, there is a statement to verify instead of a question to answer. There is never a need to analyze the actual layout, as both assume comprehension of a provided data structure in a way that is equivalent to a database table. Consequently, the methods proposed here are distinct from those used in Document Understanding [39, 7].

## 2.2 Gaps and mistakes in Document Understanding evaluation

Currently available datasets and previous work in the field cannot on their own provide enough information that would allow researchers to generalize results to other tasks within the Document Understanding paradigm. It is crucial to validate models on many tasks with a variety of characteristics a Document Understanding system may encounter in real-world applications. Notably, the scope of the challenges in a single dataset is limited to a specific task (e.g., Key Information Extraction, Question Answering) or to a particular (sub)problem (e.g., processing long documents in Kleister [45], layout understanding in DocBank [30]).

Simultaneously, a common practice in the community is to evaluate models on private data [27, 12, 37, 31] or task-specific datasets selected by authors independently [55, 56, 63, 40, 1, 19], making fair comparison difficult. Many publicly available datasets are too small to enable reliable comparison (FUNSD [24], Kleister NDA [45]) or are almost solved, i.e., there is no room for improvement due to annotation errors and near-perfect scores achieved by models nowadays (SROIE [21], CORD [38], RVL-CDIP [17]).

In light of the above circumstances, the review and selection of representative and reliable tasks is of great importance.

## 3 End-to-end Document Understanding benchmark

The primary motivation for proposing this benchmark was to select datasets covering the broad range of tasks and DU-related problems satisfying the highest quality, difficulty, and licensing criteria.

Importantly, we opt for an end-to-end nature of tasks as opposed to, e.g., problems assuming some prior information on document layout. In particular, there is no structured representation of the underlying text, such as a database-like table given in advance, and it has to be determined from the raw input file as part of the end-to-end process.

We consider the aforementioned principle of end-to-end nature crucial because it ensures measurement to which degree manual workers can be supported in their repetitive tasks, i.e., how the ultimate goal of document understanding systems is supported in real-world applications. The said *alignment with real applications* is a vital characteristic of a good benchmark [29, 43].

### 3.1 Selected datasets

Extensive documentation of the selection process, including the datasheet, is available in Appendices A-H and in the supplementary materials. Table 1 summarizes the selected tasks described in detail below, whereas Appendix A covers the complete list of considered datasets and reasons we omitted them.

Lack of the classification, layout analysis and figure QA tasks in this selection results from the fact that none of the available sets fulfills the assumed selection criteria.

Table 1: Comparison of selected datasets with their base characteristics, including information regarding whether an input is an entire document (Doc.) or document excerpt (Exc.)

| Task | Size (k documents) | | | Mean samples per document | Type | Metric | Features | | Domain |
|------|------|------|------|------|------|------|------|------|------|
| | Train | Dev | Test | | | | Input | Scanned | |
| DocVQA | 10.2 | 1.3 | 1.3 | 3.9 | Visual QA | ANLS | Doc. | + | Business |
| InfographicsVQA | 4.4 | 0.5 | 0.6 | 5.5 | Visual QA | ANLS | | − | Open |
| Kleister Charity | 1.7 | 0.4 | 0.6 | 7.8 | KIE | F1 | | +/− | Legal |
| PWC | 0.2 | 0.06 | 0.12 | 25.5 | KIE | ANLS[2] | | − | Scientific |
| DeepForm★ | 0.7 | 0.1 | 0.3 | 4.8 | KIE | F1 | | +/− | Finances |
| WikiTableQuestions★ | 1.4 | 0.3 | 0.4 | 11.3 | Table QA | Acc. | Exc. | − | Open |
| TabFact★ | 13.2 | 1.7 | 1.7 | 7.1 | Table NLI | Acc. | | − | Open |

The ★ symbol denotes that the dataset was reformulated or modified to improve its quality or align with the Document Understanding paradigm (see Table 2 and Appendix C). This symbol is not used to distinguish minor changes, such as data deduplication introduced in multiple datasets (Appendix B).

**DocVQA.** Dataset for Question Answering over single-page excerpts from various real-world industry documents. Typical questions present here might require comprehension of images, free text, tables, lists, forms, or their combination [33]. The best-performing solutions so far make use of layout-aware multi-modal models employing either encoder-decoder or sequence labeling architectures [40, 56].

**InfographicsVQA.** The task of answering questions about visualized data from a diverse collection of infographics, where the information needed to answer a question may be conveyed by text, plots, graphical or layout elements. Currently, the best result is obtained by an encoder-decoder model [32, 40].

**Kleister Charity.** A task for extracting information about charity organizations from their published reports is considered, as it is characterized by careful manual annotation by linguists and a significant gap to human performance [45]. It addresses important areas, namely high layout variability (lack of templates), need for performing an OCR, the appearance of long documents, and multiple spatial features (e.g., tables, lists, and titles).

**PWC★.** Papers with Code Leaderboards dataset was designed to extract result tuples from machine learning papers, including information on task, dataset, metric name, score. The best performing approach involves a multi-step pipeline, with modules trained separately on identified subproblems [26]. In contrast to the original formulation, we provide a complete paper as input instead of the table. This approach allows us to treat the problem as an end-to-end Key Information Extraction task with grouped variables (Appendix C).

**DeepForm★.** KIE dataset consisting of socially important documents related to election spending. The task is to extract contract number, advertiser name, amount paid, and air dates from advertising disclosure forms submitted to the Federal Communications Commission [47]. We use a subset of distributed datasets and improve annotations errors and make the annotations between subsets for different years consistent (Appendix C).

**WikiTableQuestions (WTQ)★.** Dataset for QA over semi-structured HTML tables sourced from Wikipedia. The authors intended to provide complex questions, demanding multi-step reasoning on a series of entries in the given table, including comparison and arithmetic operations [39]. The problem is commonly approached assuming a semantic parsing paradigm, with an intermediate state of formal meaning representation, e.g., inferred query or predicted operand to apply on selected cells [58, 18]. We reformulate the task as document QA by rendering the original HTML and restrict available information to layout given by visible lines and token positions (Appendix C).

**TabFact★.** To study fact verification with semi-structured evidence over relatively clean and simple tables collected from Wikipedia, entailed and refuted statements corresponding to a single row or cell were prepared by the authors of TabFact [7]. Without being affected by the simplicity of binary classification, this task poses challenges due to the complex linguistic and symbolic reasoning

---

[2]The ANLS metric used in PWC, representing KIE with property groups, differs from one used in VQA. Since it is not known how many groups are to be returned, the basis of the metric is the F1 score (in contrast to accuracy). Moreover, we require exact math for numerical variables. See implementation in the repository.

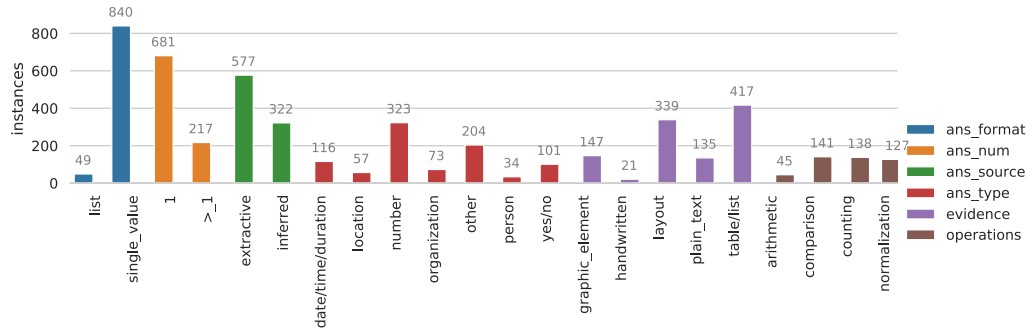

Figure 2: Number of annotated instances in each diagnostic subset category. All datasets in total.

required to perform with high accuracy. Analogously to WTQ, we render tables and reformulate the task as document NLI (Appendix C).

## 3.2 Diagnostic subsets

As pointed out by Ruder, *to better understand the strengths and weaknesses of our models, we furthermore require more fine-grained evaluation* [43]. We propose several auxiliary validation subsets, spanning across all the tasks, to improve result analysis and aid the community in identifying where to focus its efforts. A detailed description of these categories and related annotation procedures is provided in Appendix F.

**Answer characteristic.** We consider four features regarding the shallow characteristic of the answer. First, we indicate whether the answer is provided in the text explicitly in exact form (*extractive* data point) or has to be inferred from the document content (*abstractive* one). The second category includes, e.g., all the cases where value requires normalization before being returned (e.g., changing the date format). Next, we distinguish expected answers depending on whether they contain a *single value* or *list* of values. Finally, we decided to recognize several popular data types depending on shapes or class of expected named entity, i.e., to distinguish *date, number, yes/no, organization, location, and person* classes.

**Evidence form.** As we intend to analyze systems dealing with rich data, it is natural to study the performance w.r.t. the form that evidence is presented within the analyzed document. We distinguished *table/list, plain text, graphic element, layout,* and *handwritten* categories.

**Required operation.** Finally, we distinguish whether i.e., *arithmetic operation, counting, normalization* or some form of *comparison* has to be performed to answer correctly.

Table 2: Brief characteristics of our contribution, major fixes and modifications introduced to particular datasets. The enhancements of "Reformulation as DU" or "Improving data splits" are marked with ★ and are sufficient to consider the dataset unique; hence, achieved results are not comparable to the previously reported. See Appendix C for a full description of tasks processing.

| Dataset | Diagnostic sets | Unified format | Human performance | Manual annotation | Reformulation as DU | Improved split |
|---|---|---|---|---|---|---|
| DocVQA | + | + | − | − | − | − |
| InfographicsVQA | + | + | − | − | − | − |
| Kleister Charity | + | + | − | − | − | − |
| PWC★ | + | + | + | + | + | + |
| DeepForm★ | + | + | + | + | − | + |
| WikiTableQuestions★ | + | + | + | − | + | + |
| TabFact★ | + | + | − | − | + | − |

Datasets included in the benchmark differ in task type, origin, and answer form. As their random samples were annotated, diagnostic categories are not distributed uniformly and reflect the character of the problems encountered in a particular task (see Figures 10–11 in the Appendix). For example, the requirement of answer normalization is prevalent in KIE tasks of DeepForm, PWC, and Kleister Charity but not elsewhere. Consequently, the general framework of diagnostic subsets we designed can be used not only to analyze model performance but also to characterize the datasets themselves.

### 3.3 Intended use

**Data.** We propose a unified data format for storing information in the Document Understanding domain and deliver converted datasets as part of the released benchmark (all selected datasets are hosted on the `https://duebenchmark.com/data` and can be downloaded from there). It assumes three interconnected dataset, document annotation and document content levels. The dataset level is intended for storing the general metadata, e.g., name, version, license, and source. The documents annotation level is intended to store annotations available for individual documents within datasets and related metadata (e.g., external identifiers). The content level store information about output and metadata from a particular OCR engine that was used to process documents (Appendix G).

**Evaluation protocol.** To evaluate a system on the DUE benchmark, one must create a JSON file with the results (in the data format mentioned above) based on the provided test data for each dataset and then upload all of the data to the website. Moreover, we establish a set of rules (Appendix H) which guarantees that all the benchmark submissions will be fair to compare, reproducible, and transparent (e.g., training performed on a development set is not allowed).

**Leaderboard.** We provide an online platform for the evaluation of Document Understanding models. To keep an objective means of comparison with the previously published results, we decided to retain the initially formulated metrics. To calculate the global score we resort to an arithmetic mean of different metrics due to its simplicity and straightforward calculation.[3] In our platform we focus on customization, e.g., multiple leaderboards are available, and it is up to the participant to decide whether to evaluate the model on an entire benchmark or particular category. Moreover, we pay attention to the explanation by providing means to analyze the performance concerning document or problem types (e.g., using the diagnostic sets we provide).[4]

## 4   Experiments

Following the evaluation protocol, the training is run three times for each configuration of model size, architecture, and OCR engine. We performed OCR pre-processing stage for DocVQA, InfographicsVQA, Kleister Charity, and DeepForm datasets since they have PDF (mix of scans and born-digital documents) or image files as an input. PWC, WikiTableQuestions and TabFact datasets contain all born-digital documents so the ground true data are available and there is no need to run OCR engine (see Appendix C). In both cases, the pre-processing stage as an output return textual tokens and their locations (bounding boxes and page number) as a list (as a result the reading order is also provided).

### 4.1   Baselines

The focus of the experiments was to calculate baseline performance using a simple and popular model capable of solving all tasks without introducing any task-specific alterations. Employed methods were based on the previously released T5 model with a generic layout-modeling modification and pretraining.

**T5.** Text-to-text Transformer is particularly useful in studying performance on a variety of sequential tasks. We decided to rely on its extended version to identify the current level of performance on the chosen tasks and to facilitate future research by providing extendable architecture with a straightforward training procedure that can be applied to all of the proposed tasks in an end-to-end manner [41].

---

[3]Scores on the DocVQA and InfographicsVQA test sets are calculated using the official website.

[4]We intend to gather datasets not included in the present version of the benchmark to facilitate evaluations in an entire field of DU, regardless of if they are included in the current version of the leaderboard.

Table 3: Best results of particular model configuration in relation to human performance and external best. The external bests marked with — were omitted due to the significant changes in the data sets. $U$ stands for unsupervised pretraining.

| Dataset / Task type | \multicolumn{6}{c}{Score (task-specific metric)} | | | | | |
|---|---|---|---|---|---|---|
| | T5 | T5+2D | T5+U | T5+2D+U | External best | Human |
| DocVQA | $70.4_{\pm2.1}$ | $69.8_{\pm0.7}$ | $76.3_{\pm0.3}$ | $81.0_{\pm0.2}$ | 87.1 [40] | 98.1 |
| InfographicsVQA | $36.7_{\pm0.6}$ | $39.2_{\pm1.0}$ | $37.1_{\pm0.2}$ | $46.1_{\pm0.1}$ | 61.2 [40] | 98.0 |
| Kleister Charity | $74.3_{\pm0.3}$ | $72.6_{\pm1.1}$ | $76.0_{\pm0.1}$ | $75.9_{\pm0.7}$ | 83.6 [63] | 97.5 |
| PWC★ | $25.3_{\pm3.3}$ | $25.7_{\pm1.0}$ | $27.6_{\pm0.6}$ | $26.8_{\pm1.8}$ | — | 69.3 |
| DeepForm★ | $74.4_{\pm0.6}$ | $74.0_{\pm0.7}$ | $82.9_{\pm0.9}$ | $83.3_{\pm0.3}$ | — | 98.5 |
| WikiTableQuestions★ | $33.3_{\pm0.7}$ | $30.8_{\pm1.9}$ | $38.1_{\pm0.1}$ | $43.3_{\pm0.4}$ | — | 76.7 |
| TabFact★ | $58.9_{\pm0.5}$ | $58.0_{\pm0.3}$ | $76.0_{\pm0.1}$ | $78.6_{\pm0.1}$ | — | 92.1 |
| Visual QA | 53.6 | 54.5 | 56.7 | 63.5 | n/a | 98.1 |
| KIE | 69.1 | 67.7 | 74.8 | 76.4 | n/a | 88.4 |
| Table QA/NLI | 29.4 | 29.0 | 38.0 | 39.3 | n/a | 84.4 |
| Overall | 50.7 | 50.4 | 56.5 | 59.8 | n/a | 90.3 |

**T5+2D.** Extension of the model we propose assumes the introduction of 2D positional bias that has been shown to perform well on tasks that demand layout understanding [56, 40, 63]. We rely on 2D bias in a form introduced in TILT model [40] and provide its first open-source implementation (available in supplementary materials). We expect that comprehension of spatial relationships achieved in this way will be sufficient to demonstrate that methods from the plain-text NLP can be easily outperformed in the DUE benchmark.

**Unsupervised pretraining.** We constructed a corpus of documents with a visually rich structure, based on 480k PDF files from the UCSF Industry Documents Library. It is used with a T5-like masked language model pretraining objective but in a salient span masking scheme where named entities are preferred over random tokens [41, 15]. An expected gain from its use is to tune 2D biases and become more robust to OCR errors and incorrect reading order.[5]

**Human performance.** We relied on the original estimation for DocVQA, InfographicsVQA, Charity, and TabFact datasets. For the PWC, WTQ and DeepForm estimation of human performance, we used the help of professional in-house annotators who are full-time employees of our company (see Appendix E). Each dataset was handled by two annotators; the average of their scores, when validated against the gold standard, is treated as the human performance (see Table 3). Interestingly, human scores on PWC are relatively low in terms of ANLS value – we explained this and justified keeping the task in Appendix C.

## 4.2 Results

Comparison of the best-performing baselines to human performance and top results reported in the literature is presented in Table 3. In several cases, there is a small difference between the performance of our baselines and the external best. It can be attributed to several factors. First, the best results previously obtained on the tasks were task-specific, i.e., were explicitly designed for a particular task and did not support processing other datasets within the benchmark. Secondly, there are differences between the evaluation protocol that we assume and what the previous authors assumed (e.g., we do not allow training models on the development sets, we require reporting an average of multiple runs, we disallow pretraining on datasets that might lead to information leak). Thirdly, our baseline could not address examples demanding vision comprehension as it does not process image inputs. Finally, there is the case of Kleister Charity. An encoder-decoder model we relied on as a one-to-fit-all baseline cannot process an entire document due to memory limitations. As a result, the score was lower as we consumed only a part of the document. Note that external bests for reformulated tasks are no longer applicable to the benchmark in its present, more demanding form.

---

[5]Details of the training procedure, such as used hyperparameters and source code, are available in the repository accompanying the paper.

Irrespective of the task and whether our competitive baselines or external results are considered, there is still a large gap to humans, which is desired for novel baselines. Moreover, one can notice that the addition of 2D positional bias to the T5 architecture leads to better scores assuming the prior pretraining step, which is yet another result we anticipated as it suggests that considered tasks have an essential component of layout comprehension.

Interestingly, the performance of the model can be significantly enhanced (up to 20.6 points difference for TabFact dataset and T5+2D+U model) by providing additional data for the said unsupervised pretraining. Thus, the results not only support the premise that understanding 2D features demand more unlabeled data than the chosen datasets can offer but also lay a common ground between them, as the same layout-specific pretraining improved performance on all of them independently. This observation confirms that the notion of layout is a vital part of the chosen datasets.

### 4.3 Challenges of the Document Understanding domain

Owing to its end-to-end nature and heterogeneity, Document Understanding is the touchstone of Machine Learning. However, the challenges begin to pile up due to the mere form a document is available in, as there is a widespread presence of analog materials such as scanned paper records. In the analysis below, we aim to explore the field of DU from the perspective of the model's development and point out the most critical limiting factors for achieving satisfying results.

**Impact of OCR quality.** We present detailed results for Azure CV and Tesseract OCR engine in Table 5. The differences in scores are huge for most of the datasets (up to $18.4\%$ in DocVQA) with a clean advantage for Azure CV. Consequently, we see that architectures evaluated with different OCR engines are incomparable, e.g., the choice of an OCR engine may impact results more than the choice of model architecture. Moreover, with the usage of our diagnostic datasets we can observe that Tesseract struggle the most with *Handwritten* and *Table/list* categories in comparison to *Plain text* category. It is worth noting that we see a bigger difference in the results between Azure CV and Tesseract for *Extractive* category, which suggest that we should use better OCR engines especially for that kind of problems.

**Requirement of multi-modal comprehension.** In addition to layout and textual semantics, part of the covered problems demand a Computer Vision component, e.g., to detect a logo, analyze a figure, recognize text style, determine whether the document was signed or the checkbox nearby was selected. This has been confirmed by ablation studies performed by Powalski et al. [40] for the DocVQA and by the fact that models with vision component achieve better performance on leaderboards for datasets such as DocVQA and the InfographicsVQA datasets [40, 56, 23, 22]. Thus, Document Understanding naturally incorporates challenges of both multi-modality and each modality individually (but not for all tasks equally, see Figures 10–11 in the Appendix). Since none of our baselines contain a vision component, we underperform on the category of problems requiring multi-modality, as is visible on the diagnostic dataset we proposed. Nevertheless, better performance of the T5+2D model suggests that part of the problems considered as *visual*, can be to some extent approximated by solely using the words' spatial relationships (e.g., text curved around a circle, located in the top-left corner of the page presumably has the logo inside).

**Single architecture for all datasets.** It is common that token-level annotation is not available, and one receives merely key-value or question-answer pairs assigned to the document. Even in problems of extractive nature, token spans cannot be easily obtained, and consequently, the application of state-of-the-art architectures from other tasks is not straightforward. In particular, authors attempting Document Understanding problems in sequence labeling paradigms were forced to rely on faulty handcrafted heuristics [40]. In the case of our baseline models, this problem is addressed straight-forwardly by assuming a sequence-to-sequence paradigm that does not make use of token-level annotation. This solution, however, comes with a trade-off of low performance on datasets requiring comprehension of long documents, such as Kleister Charity (see Table 4).

Table 4: F1 score on the Kleister Charity challenge with various maximum input sequence lengths.

| Dataset | Maximum input sequence length | | | |
|---|---|---|---|---|
| | 1024 | 2048 | 4096 | 6144 (max) |
| Kleister Charity | 56.6 | 66 | 73.2 | 75.9 |

Table 5: Scores for different OCR engines and datasets with T5+2D model performing on 1024 tokens.

| OCR | DocVQA | IVQA | Charity | DeepForm | Average | Average scores for different diagnostic categories | | | | |
| --- | --- | --- | --- | --- | --- | --- | --- | --- | --- | --- |
| | | | | | | Extractive | Inferred | Handwritten | Table/list | Plain text |
| Azure CV (v3.2) | 71.8 | 40.0 | 57.7 | 74.8 | 61.1 | 51.3 | 33.0 | 31.3 | 46.0 | 65.3 |
| Tesseract (v4.0) | 55.7 | 28.3 | 55.7 | 66.8 | 51.6 | 43.1 | 29.5 | 12.5 | 27.2 | 61.1 |

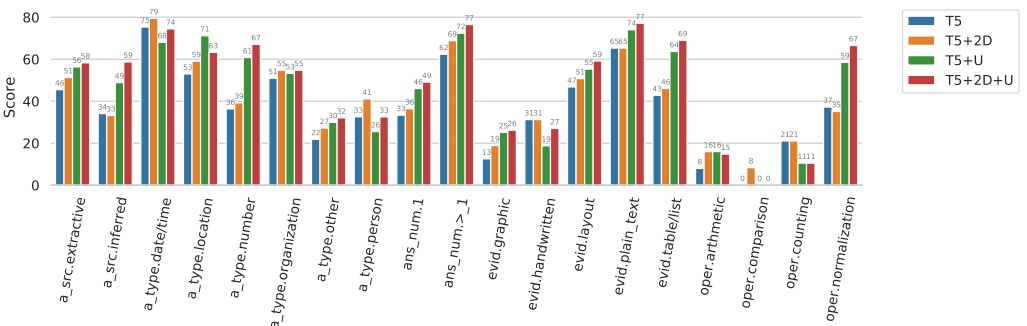

Figure 3: Results for diagnostic subsets. See Appendix F for detailed description of these categories.

**Diagnostic dataset.** Our diagnostic datasets are an important part of the analysis of different challenges in general (e.g., OCR quality or multi-modal comprehension, as we mentioned above) and for debugging different types of architectural decisions (see Figure 3). For example, we can observe a big advantage of unsupervised pretraining in the *inferred, number, table/list* categories, which shows the importance of a good dataset for specific problems (dataset used for pretraining the original T5 model has a small number of documents containing tables). The most problematic categories for all models were those related to complex logic operations: *arithmetic, counting, comparison*.

## 5   Conclusions

To efficiently pass information to the reader, writers often assume that structured forms such as tables, graphs, or infographics are more accessible than sequential text due to human visual perception and our ability to understand a text's spatial surroundings. We investigate the problem of correctly measuring the progress of models able to comprehend such complex documents and propose a benchmark – a suite of tasks that balance factors such as quality of a document, importance of layout information, type and source of documents, task goal, and the potential usability in modern applications.

We aim to track the future progress on them with the website prepared for transparent verification and analysis of the results. The former is facilitated by the diagnostics subsets we derived to measure vital features of the Document Understanding systems. Finally, we provide a set of solid baselines, datasets in the unified format, and released source code to bootstrap the research on the topic.

## Acknowledgments and disclosure of funding

The authors would like to thank Samuel Bowman, Łukasz Garncarek, Dimosthenis Karatzas, Minesh Mathew, Zofia Prochoroff, and Rubèn Pérez Tito for the helpful discussions on the draft of the paper. Moreover, we thank the reviewers of both rounds of the NeurIPS 2021 Datasets and Benchmarks Track for their comments and suggestions that helped improve the paper.

The Smart Growth Operational Programme supported this research under project no. POIR.01.01.01-00-0877/19-00 (*A universal platform for robotic automation of processes requiring text comprehension, with a unique level of implementation and service automation*).

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
