# A  Considered datasets

## A.1  Desired characteristics

**End-to-end nature.**  As the value and importance of Document Understanding result from its application to process automation, a good benchmark should measure to which degree workers can be supported in their tasks. Though Layout Analysis is oldest of the Document Understanding problems, its output is often not an end in itself but rather a half-measure disconnected from the final information the system is used for. We also remove all tasks which as an input takes collection of documents.

**Quality.**  Availability of high-quality annotation was a condition *sine qua non* for a task to qualify. To ensure the highest annotation quality, we excluded resources prepared using a distant annotation procedure, e.g., classification tasks where entire sources were labeled instead of individual instances, or templated question-answer pairs.

**Difficulty.**  As it makes no sense to measure progress on solved problems, only tasks with a substantial gap between human performance and state-of-the-art models were considered. In the case of promising tasks lacking a human baseline, we provided our estimation. Moreover, we remove all tasks were free text was dominated in documents (we don't need to use layout or visual features).

**Licensing.**  In publishing our benchmark, we are making efforts to ensure the highest standards for the future of the machine learning community. Only tasks with a permissive license to use annotations and data for further research can be considered.

At the same time, we recognized it is essential to approach the benchmark construction holistically, i.e., to carefully select tasks from diverse domains and types in the rare cases where datasets are abundant.

## A.2  Datasets selection process

The review protocol consisted of a manual search in specific databases, repositories and distribution services. The scientific resources included in the search were:

- `https://paperswithcode.com/datasets/`
- `https://datasetsearch.research.google.com/`
- `https://data.mendeley.com/`
- `https://arxiv.org/search/`
- `https://github.com/`
- `https://allenai.org/data/`
- `https://www.semanticscholar.org/`
- `https://scholar.google.com/`
- `https://academic.microsoft.com/home`

Results were reviewed by one of authors of the present paper and the resources related to classification, KIE, QA, MRC, and NLI over complex documents, figures, and tables were identified as potentially relevant (in accordance with inclusion criteria described in Section A.1).

The initial search assumed use of the following keywords: *Question Answering, Visual Question Answering, Document Question Answering, Document Classification, Document Dataset, Information Extraction*. Additionally, we used *Machine Reading Comprehension, Question Answering, VQA* in combination with *Document*, and *Visual, Document, Table, Figure, Plot, Chart, Hybrid* in combination with *Question Answering* or *Information Extraction*.

Table 6 presents list of relevant datasets and results of their assessment according to the criteria of end-to-end nature, quality, difficulty, and licensing. Candidate tasks resulted from an extensive review of both literature and data science challenges without accompanying publication and their basic characteristics.

Table 6: Comparison of selected and considered datasets with their base characteristic, including information regarding whether an input is a collection of documents (Col.), entire document (Doc.) or document excerpt (Exc.).

| Dataset | Type | Size (thousands) | | | Selection criteria | | | | Input | Domain | Comment |
|---|---|---|---|---|---|---|---|---|---|---|---|
| | | Train | Dev | Test | End-to-end | Quality | Difficulty | Licensing | | | |
| Kleister Charity [45] | KIE | 1.73 | .44 | .61 | + | + | + | + | Doc. | Finances | |
| PWC [26] | KIE | .2 | .06 | .12 | + | + | + | + | Doc. | Scientific | |
| DeepForm [47] | KIE | .7 | .1 | .3 | + | + | + | + | Doc. | Finances | |
| DocVQA [33] | Visual QA | 10.2 | 1.3 | 1.3 | + | + | + | + | Doc. | Business | |
| InfographicsVQA [32] | Visual QA | 4.4 | .5 | .6 | + | + | + | + | Doc. | Open | |
| TabFact [7] | Table NLI | 13.2 | 1.7 | 1.7 | + | + | + | + | Exc. | Open | |
| WTQ [39] | Table QA | 1.4 | .3 | .4 | + | + | + | + | Exc. | Open | |
| Kleister NDA [45] | KIE | .25 | .08 | .2 | + | + | − | + | Doc. | Legal | Dominated by extraction from free text |
| SROIE [20] | KIE | .63 | - | .35 | + | + | − | + | Doc. | Finances | No room for improvement |
| CORD [38] | KIE | .8 | .1 | .1 | + | + | − | + | Doc. | Finances | No room for improvement |
| Wildreceipt [46] | KIE | 1.27 | - | .47 | + | + | − | + | Doc. | Finances | No room for improvement |
| WebSRC [5] | KIE | 4.55 | .9 | 1.0 | + | − | + | + | Doc. | Open | Templated input data |
| FUNSD [24] | KIE | .15 | - | .05 | + | + | − | + | Doc. | Finances | Known disadvantages [51] |
| DocVQA [32] | Visual QA | 4.4 | .5 | .6 | − | + | + | + | Col. | Open | Document Collection Question Answering |
| TextbookQA [28] | Visual QA | .67 | .2 | .21 | + | − | + | + | Doc. | Educational | Source files are not available |
| MultiModalQA [48] | Visual QA | 23.82 | 2.44 | 3.66 | + | − | + | + | Doc. | Open | Automatically generated questions |
| VisualMRC [49] | Visual MRC | 7 | 1 | 2 | + | + | − | + | Doc. | Open | Human performance reached |
| RVL-CDIP [17] | Classification | 320 | 40 | 40 | + | + | − | + | Doc. | Finances | No room for improvement |
| DocFigure [25] | Classification | 19.8 | - | 13.1 | + | + | − | + | Doc. | Scientific | No room for improvement |
| EURLEX57K [3] | Classification | 45 | 6 | 6 | + | + | − | + | Doc. | Legal | Dominated by extraction from free text |
| MELINDA [54] | Classification | 4.34 | .45 | .58 | + | − | + | + | Doc. | Scientific | Semi-supervised annotation |
| S2-VL [44] | DLA | 1.3 | - | - | − | − | + | + | Doc. | Scientific | Cross-validation for training and testing |
| DocBank [30] | DLA | 398 | 50 | 50 | − | − | + | + | Doc. | Scientific | Automatic annotation |
| Publaynet [61] | DLA | 340.4 | 11.9 | 12 | − | − | + | + | Doc. | Scientific | Automatic annotation |
| FinTabNet [60] | DLA | 61.8 | 7.19 | 7.01 | − | + | + | + | Doc. | Finances | Different styles in comparison to sci./gov. docs |
| PlotQA [34] | Figure QA | 157 | 33.7 | 33.7 | + | − | + | + | Exc. | Open | Synthetic |
| Leaf-QA [4] | Figure QA | 200 | 40 | 8.15 | + | − | + | + | Exc. | Open | Templated questions |
| TAT-QA [62] | Table QA | 2.2 | .28 | .28 | + | − | + | + | Exc. | Finances | Source files are not available |
| WikiOPS [9] | Table QA | 17.28 | 2.47 | 4.67 | + | + | − | + | Exc. | Open | No room for improvement |
| FeTaQA [35] | Table QA | 7.33 | 1.0 | 2.0 | + | − | + | + | Exc. | Open | Answers as a free-form text |
| HybridQA [8] | Table QA | 62.68 | 3.47 | 3.46 | − | + | + | + | Col. | Open | Multihop Question Answering |
| OTT-QA [6] | Table QA | 41.46 | 2.24 | 2.16 | − | + | + | + | Col. | Open | Multihop Question Answering |
| INFOTABS [14] | Table NLI | 1.74 | .2 | .6 | + | + | + | + | Col. | Open | TabFact is very similar |

# B   Minor dataset modifications

**Deduplication.**   Through the systematic analysis and validation of the chosen datasets, we noticed one of the commonly appearing defects is the presence of duplicated annotations. We decided to remove these duplicates from InfographicsVQA (14 annotations from train, two from the dev set), DocVQA (four from train and test sets each), TabFact (309 from train, 53 from dev, and 52 the test set), and WikiTableQuestions (one annotation from each train and test sets).

# C   Tasks processing and reformulation

Since part of the datasets were reformulated or modified to improve the benchmark quality or align the task with the Document Understanding paradigm, we describe the introduced changes in detail below.

**WikiTableQuestions⋆.**   We prepare input documents by rendering table-related HTML distributed by authors in *wkhtmltopdf* and crop the resulting files with *pdfcrop*. As these code excerpts do not contain *head* tag with JavaScript and stylesheet references, we use the header from the present version of the Wikipedia website.

Approximately 10% of tables contained at least one *img* tag with a source that is no longer reachable. It results in a question mark icon displayed instead of the image and does not impact the evaluation procedure since the questions here do not require image comprehension.

The original WTQ dataset consists of *training*, *pristine-seen-tables*, and *pristine-unseen-tables* subsets. We treat *pristine-unseen-tables* as a test set and create new training and development sets by rearranging data from *training* and *pristine-seen-tables*. The latter operation is dictated by the leakage of documents in the original formulation, i.e., we consider it undesirable for a document to appear in different splits, even if the question differs. The resulting dataset consists of approximately

| Year | Venue | Winners | Runner-up | 3rd place |
|------|-------|---------|-----------|-----------|
| 2005 | Pardubice | Poland (41 pts) | Sweden (35 pts) | Denmark (24 pts) |
| 2006 | Rybnik | Poland (41 pts) | Sweden (27 pts) | Denmark (26 pts) |
| 2007 | Abensberg | Poland (40 pts) | Great Britain (36 pts) | Czech Republic (30 pts) |
| 2008 | Holsted | Poland (40 pts) | Denmark (39 pts) | Sweden (38 pts) |
| 2009 | Gorzów Wlkp. | Poland (57 pts) | Denmark (45 pts) | Sweden (32 pts) |
| 2010 | Rye House | Denmark (51 pts) | Sweden (37 pts) | Poland (35 pts) |
| 2011 | Balakovo | Russia (61 pts) | Denmark (31 pts) | Ukraine (29+3 pts) |
| 2012 | Gniezno | Poland (61 pts) | Australia (44 pts) | Sweden (26 pts) |
| Year | Venue | Winners | Runner-up | 3rd place |

Figure 4: Document in WikiTableQuestions reformulated as Document Understanding.

(Question) After their first place win in 2009, how did Poland place the next year at the speedway junior world championship? (Answer) 3rd place

2100 documents divided in the proportion of 65%, 15%, 20% into training, development, and test sets.

We rely on the original WTQ metric which is a form of Accuracy with normalization (see Pasupat et al. [39] and accompanying implementation).

**TabFact⋆.** As the authors of TabFact distribute only CSV files, we resorted to HTML from the WikiTables dump their CSV were presumably generated from.[6] As Chen et al. [7] dropped some of the columns present in used WikiTable tables, we remove them, to ensure compatibility with the original TabFact. Rendered files are used analogously to the case of WTQ.

Superleague (Final League) Table (Places 1-6)

| | Nation | v t e Games | | | | Points | | | Table points |
|---|--------|--------|-----|-------|------|-----|---------|------------|--------------|
| | | Played | Won | Drawn | Lost | For | Against | Difference | |
| 1 | VVA-Podmoskovye Monino | 10 | 9 | 0 | 1 | 374 | 119 | +255 | 37 |
| 2 | Krasny Yar Krasnoyarsk | 10 | 6 | 0 | 4 | 198 | 255 | -57 | 28 |
| 3 | Slava Moscow | 10 | 5 | 1 | 4 | 211 | 226 | -15 | 26 |
| 4 | Yenisey-STM Krasnoyarsk | 10 | 5 | 0 | 5 | 257 | 158 | +99 | 25 |
| 5 | RC Novokuznetsk | 10 | 4 | 1 | 5 | 168 | 194 | -26 | 23 |
| 6 | Imperia-Dynamo Penza | 10 | 0 | 0 | 10 | 138 | 395 | -257 | 10 |

Figure 5: Document in TabFact reformulated as Document Understanding.

(Claim) To calculate table point, a win be worth 3, a tie be worth 1 and a loss be worth 0

Results differ from TabFact in several aspects, i.e., text in our variant is not normalized, it includes the original formatting, and the tables are more complex due to restoring the original cell merges. All mentioned differences are desired, as we intended to consider raw, unprocessed files without any heuristics or normalization applied.

Another difference we noticed is that tables in the original TabFact are sometimes one row shorter, i.e., they do not contain the last row present in the WikiTable dump. As it should not impact expected answers, we decided to maintain the fidelity to Wikipedia and use the complete table.

We use the original splits into training, development, and test sets and the original Accuracy metric.

**DeepForm⋆.** The original DeepForm dataset consists of 2012, 2014, and 2020 subsets differing in terms of annotation quality and documents' diversity. We decided to use only the 2020 subset as for 2014, and 2020 annotations were prepared either automatically or by volunteers, leading to questionable quality. The selected subset was randomly divided into training, development and test set.

We noticed several inconsistencies during the initial analysis that lead us to the manual correction of autodetected: (1) invalid date format; (2) flight start dates earlier than flight end; (3) documents lacking one or more data points.

In addition to the improved 2020 subset, we manually annotated one hundred 2012 documents, as they can pose different challenges (contain different document templates, handwriting, have lower

---

[6] http://websail-fe.cs.northwestern.edu/TabEL/tables.json.gz

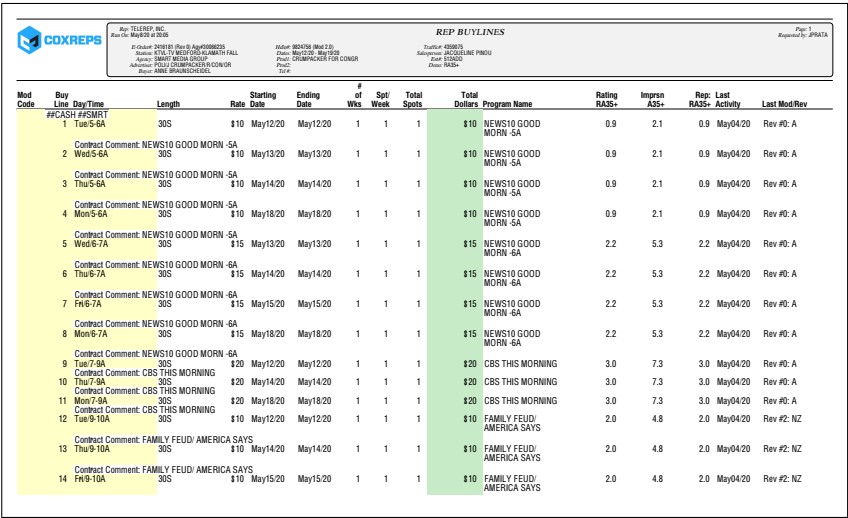

Figure 6: Single page from document in DeepForm.

image quality). They were used to extend development and test set. The final dataset consists of 700 training, 100 development, and 300 test set documents. We rely on the standard F1 score for the purposes of DeepForm evaluation.

**PWC★.** The authors of AxCell relied on PWC Leaderboards and LinkedResults datasets [26]. The original formulation assumes extraction of *(task, dataset, metric, model, score)* tuples from a provided table. In contrast, we reformulate the task as Document Understanding and provide a complete paper as input instead. These are obtained using arXiv identifiers available in the PWC metadata. Consequently, the resulting task is an end-to-end Key Information Extraction from real-world scientific documents.

Whereas LinkedResults was annotated consistently, the PWC is of questionable quality as it was obtained from leaderboards filled by Papers with Code visitors without a clear guideline or annotation rules. The difference between the two is substantial, i.e., the agreement in terms of F1 score between publications present in both PWC and LinkedResults is lower than $0.35$. We attribute this mainly to flaws in the PWC dataset, such as missing records, inconsistent normalization and the difficulty of the task itself.

Consequently, we decided to perform its manual re-annotation assuming that: (1) The best result for a proposed model variant on the single dataset has to be annotated, e.g., if two models with different parameter sizes were present in the table, we report only the best one. (2) Single number is preferred (we take the average over multiple split or parts of the dataset if possible). (3) When results from the test set are available, we prefer them and don't report results from the validation set. (4) We add multiple value variants when possible. (5) We include information on used validation/dev/test split in the dataset description wherever applicable. (6) We don't report results on the train set. (7) We don't annotate results not appearing in the table. (8) We filter out publications that are hard to annotate even for a human.

Interestingly, human scores on PWC are relatively low in terms of ANLS value. This can be attributed to unrestricted nature of particular properties, e.g., *accuracy* and *average accuracy* are equally valid metric values. Similarly, *Action Recognition*, *Action Classification*, and *Action Recognition* are equally valid task names. We mitigated this problem by using ANLS-like comparison used in the F1 metric and providing multiple acceptable value variants, i.e., it is enough to provide half of the string representing one of the valid answers.[7]

Nevertheless, it is impossible to provide all answer variants during the preparation of the gold standard. We decided to keep the dataset in the benchmark as it is extremely demanding, and there is still a large gap between humans' and models' performance (See Table 3).

---

[7]Please refer to the metric implementation in the Github repository for a detailed description.

As the expected answer in PWC consists of a list of groups (property tuples that represent a complete record of the method, dataset, and results), the F1 metric here has to take into account the miss-placement of properties in another group. We assume the value is incorrect if placed in the wrong group (see reference implementation in supplementary materials).

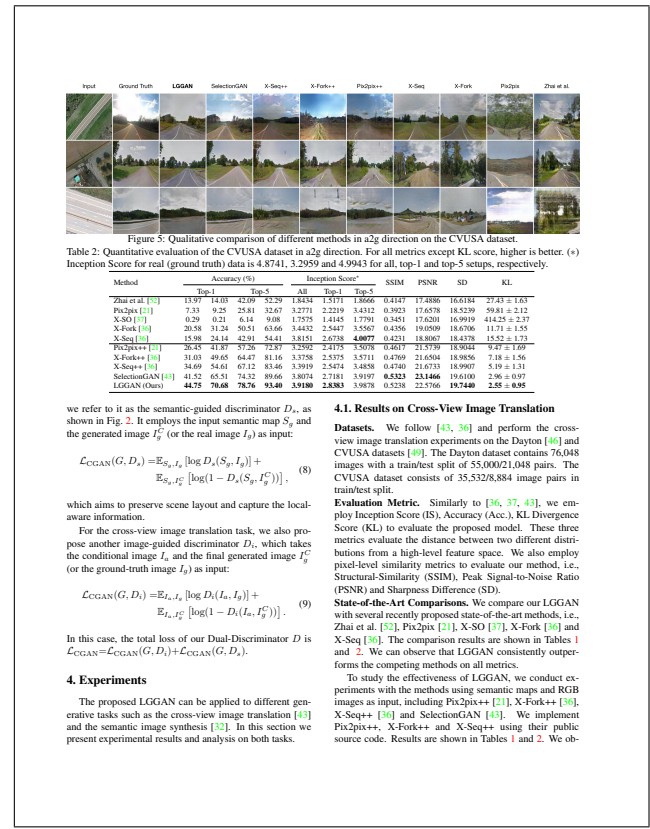

Figure 7: Single page from document in PWC.

# D   Dataset statistics

Chosen datasets represent the plethora of domains, lengths, and document types. This appendix covers the critical aspects of particular tasks at the population level.

Though part of the datasets is limited to one-pagers, the remaining documents range from a few to few hundred pages (Figure 8). At the same time, there is a great variety in how much text is present on a single page – we have both densely packed scientific documents and concise document excerpts or infographics. This diversity allows us to measure the ability to comprehend documents depending on their length.

# E   Details of human performance estimation

Estimation of human performance for PWC, WikiTableQuestions, DeepForm was performed in-house by professional annotators who are full-time employees of Applica.ai. Before approaching the process, each of them has to participate in the task-specific training described below.

Number of annotated samples depended on task difficulty and the variance of the resulting scores. We relied on 50 fully annotated papers for the PWC dataset (approx. 150 tuples with five values each), 109 DeepForm documents (532 values), and 300 questions asked to different WikiTableQuestion tables.

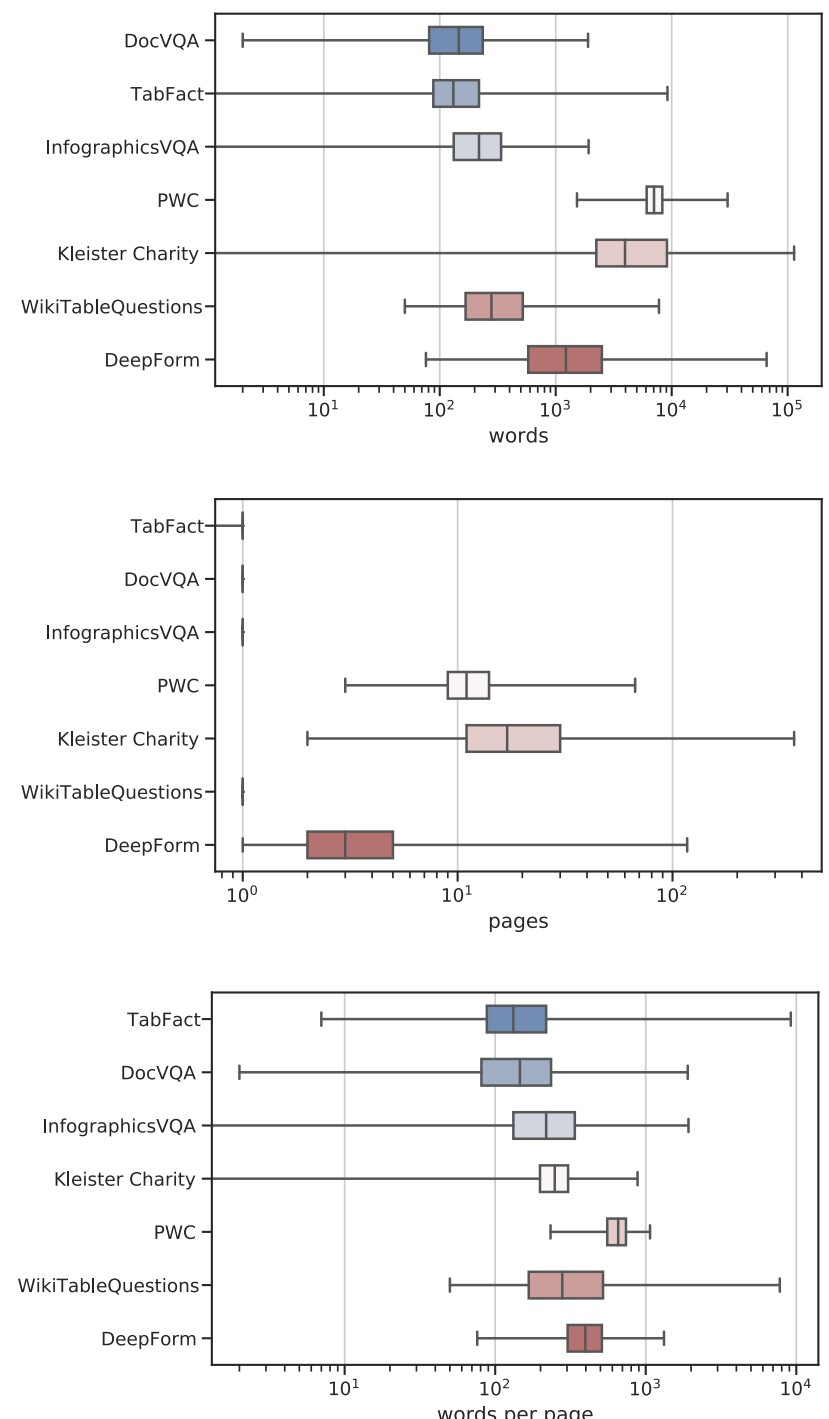

Figure 8: Number of words, pages, and words per page in particular datasets (log scale). Part of the datasets consist only of one-pagers.

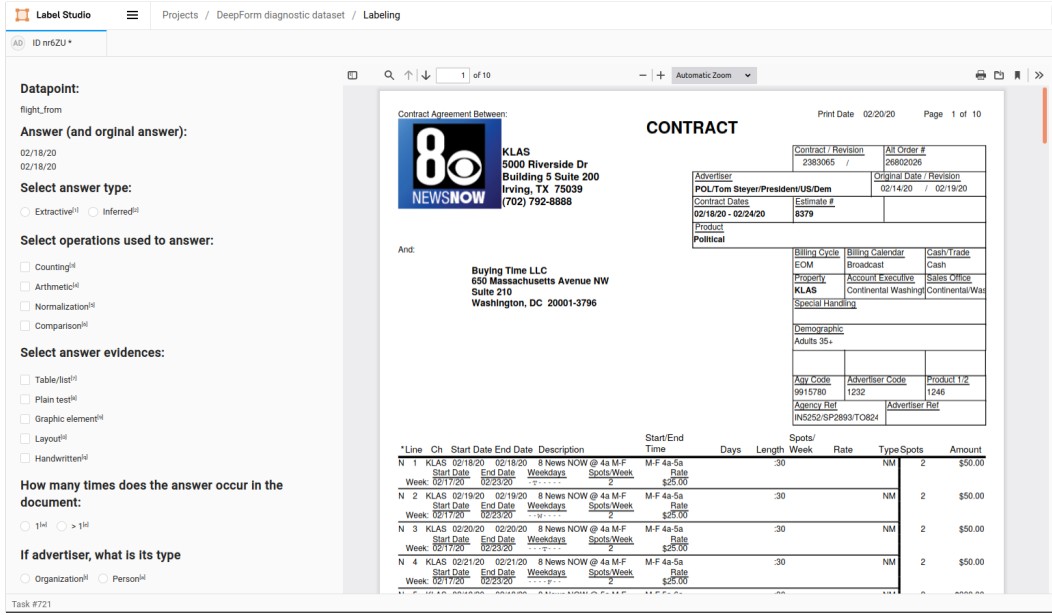

Figure 9: An example of an interface for annotating diagnostic subsets based on document from DeepForm dataset.

Each dataset was approached with two annotators in the LabelStudio tool. Human performance is the average of their scores when validated against the gold standard.

**Training.** Each person participating in the annotation process completed the training consisting of four stages: (1) Annotation of five random documents from the task-specific development set. (2) Comparative analysis of differences between their annotations and the gold standard. (3) Annotation of ten random documents from the task-specific development set and subsequent comparative analysis. (4) Discussion between annotators aimed at agreeing on the shared, coherent annotation rules.

## F Annotation of diagnostic subsets

In order to analyze the prepared benchmark and the results of individual models, diagnostic sets were prepared. These diagnostic sets are subsets of examples selected from the testset for all datasets.

When building a taxonomy for diagnostic sets, we adopted two basic assumptions: (1) It must be consistent across all selected tasks so that at least two tasks can be noted with a given category (2) It should include as many aspects as possible that are relevant from the perspective of document understanding problem.

Initially, we adopted the taxonomies proposed in DocVQA, Infographics, and TabFact as potential categories [33, 32, 7]. In the next step, we adjusted our taxonomy to all datasets following the previously adopted assumptions, distinguishing seven main categories with 25 subcategories (for a more detailed description of the category (see the section F.1). Then, for each dataset, we prepared an annotation task in the LabelStudio tool [8] (see example 9) along with an annotation instruction. Finally, to determine Human performance, the annotation was carried out by a team of specialists from Applica.ai, where the selected example was noted only by one person.

### F.1 Taxonomy description

The taxonomy is based on multiple aspects of documents, inputs, and answers and was designed to be sufficiently generic for future adaptation to other tasks. Here, in each category, we describe the predicates that annotators followed when classified an example into specific subcategories.

---

[8]https://labelstud.io/

**Answer source.** This category is based on the relation between answer and text in the document.

- Extractive – after lowercasing and white-characters removing, the answer can be exact-matched in the document.
- Inferred – other non-extractive cases.

**Output format** This category is based on the shape of an output.

- Single value – the answer consists of only one item.
- List – multiple outputs are to be provided.

**Output type.** This category is based on the semantic of an output.

- Organization – the answer is a name of an organization or institution.
- Location – the answer is a geographic location globally (e.g., a country, continent, city) or locally (building or street, among others).
- Person – the answer is a personal identifier(name, surname, pseudonym) or its composition. It can have a title prefix or suffix (e.g., Mrs., Mr., Ph.D.) or have a shortened or informal version.
- Number – numerical values given with the unit or percent. Values written in the free text do not comply with this class's definition.
- Date/Time/Duration – the answer represents the date, time, or the difference between two dates or times.
- Yes/No – the answer is a textual output of binary classification, such as Yes/No pairs, and Positive/Negative, 0/1 among others.

**Evidence.** This category is based on the source of information that allows the correct answer to be generated. When there are multiple justifications based on different pieces of evidence (for example, the address is in a table and block text), it is required to select all the pieces of evidence.

- Table or List – a *table* is a fragment of the document organized into columns and rows. The distinguishing feature of the table is consistency within rows and columns (usually the same data type). Moreover, it may have a header. In that sense, the form is not a table (or at least it does not have to be). A *list* is a table degenerated into one column or row containing a header.
- Plain text – the answer is based on plain text if there is an immediate need to understand a longer fragment of the text while answering.
- Graphic element – the answer is based on graphic evidence when understanding graphically rich, non-text fragments of documents (e.g., graphics, photos, logos (non-text)) are necessary for generating a correct answer.
- Layout – it is evidence when comprehending the placement of text on the page (e.g., titles, headers, footers, forms) is needed to generate the correct answer. This type does not include tables.
- Handwritten – when the text written by hand is crucial for an answer.

**Operation.** This category is based on the type of operations that are to be performed on the document before reaching to the correct answer.

- Counting – when there is a need to count the occurrences or determine the position on the list.
- Arithmetic – when there is an arithmetic operation applied before answering, or a sequence of arithmetic operations (e.g., averaging).
- Comparison – a comparison in the sense of lesser/greater. Other procedures that a comparison operation can express (e.g., approximation) may be chosen. Here, the operation "is equal" is not a comparison since it is sufficient to match sequences without a semantic understanding.
- Normalization – when we are to return something in the document but in a different form. It may only apply to the output; we do not acknowledge this operation when it is required to normalize a question fragment to match it in the document.

**Answer number.** This category is based on the number of occurrences of an answer in the document.

- 1 – when there is one path of logical reasoning to find the correct answer in the document. We treat it as one justification for two different reasoning paths based on the same data from the document.
- > 1 – the other cases.

## G Unified format

We propose a unified format for storing information in the Document Understanding domain and deliver converted datasets as part of the released benchmark. It assumes three interconnected levels: dataset, document-annotation and document-content. Please refer to the repository for examples and formal specifications of the schemes.

**Dataset.** The dataset level is intended for storing the general metadata, e.g., name, version, license, and source. Here, the JSON-LD format based on the well-known schema.org web standard is used.[9]

**Document.** The documents annotation level is intended to store annotations available for individual documents within datasets and related metadata (e.g., external identifiers). Our format, valid for all the Document Understanding tasks, is specified using the JSON-Schema standard. This ensures that every record is well-documented and makes automatic validation possible. Additionally, to make the processing of large datasets efficient, we provide JSON Lines file for each split, thus it is possible to read one record at a time.

**Content.** As part of the original annotation or additional data we provide is related to document content (e.g., the output of a particular OCR engine), we introduce the document's content level. Similarly to the document level, we propose an adequate JSON Schema and provide the JSON Lines files in addition. PDF files with the source document accompany dataset -, document-, and content-level annotations. If the source PDF was not available, a lossless conversion was performed.

## H Evaluation protocol

**Evaluation protocol.** All the benchmark submissions are expected to conform to the following rules to guarantee fair comparison, reproducibility, and transparency:

- All results should be automatically obtainable starting from either raw PDF documents or the JSON files we provide. In particular, it is not permitted to rely on the potentially available source file that our PDFs were generated from or in-house manual annotation.
- Despite the fact that we provide an output of various OCR mechanisms wherever applicable, it is allowed to use software from outside the list. In such cases, participants are highly encouraged to donate OCR results to the community, and we declare to host them along with other variants. It is expected to provide detailed information on used software and its version.
- Any dataset can be used for unsupervised pretraining. The use of supervised pretraining is limited to datasets where there is no risk of information leakage, e.g., one cannot train models on datasets constructed from Wikipedia tables unless it is guaranteed that the same data does not appear in WikiTableQuestions and TabFact.
- It is encouraged to use datasets already publicly available or to release data used for pretraining.
- Training performed on a development set is not allowed. We assume participants select the model to submit using training loss or validation score. We do not release test sets and keep them secret by introducing a daily limit of evaluations performed on the benchmark's website.
- Although we allow submissions limited to one category, e.g., QA or KIE, complete evaluations of models that are able to comprehend all the tasks with one architecture are highly encouraged.
- Since different random initialization or data order can result in considerably higher scores, we require the bulk submission of at least three results with different random seeds.

---

[9]See `https://json-ld.org/` for information on the JSON-LD standard, and `https://developers.google.com/search/docs/data-types/dataset` for the description of adapted schema.

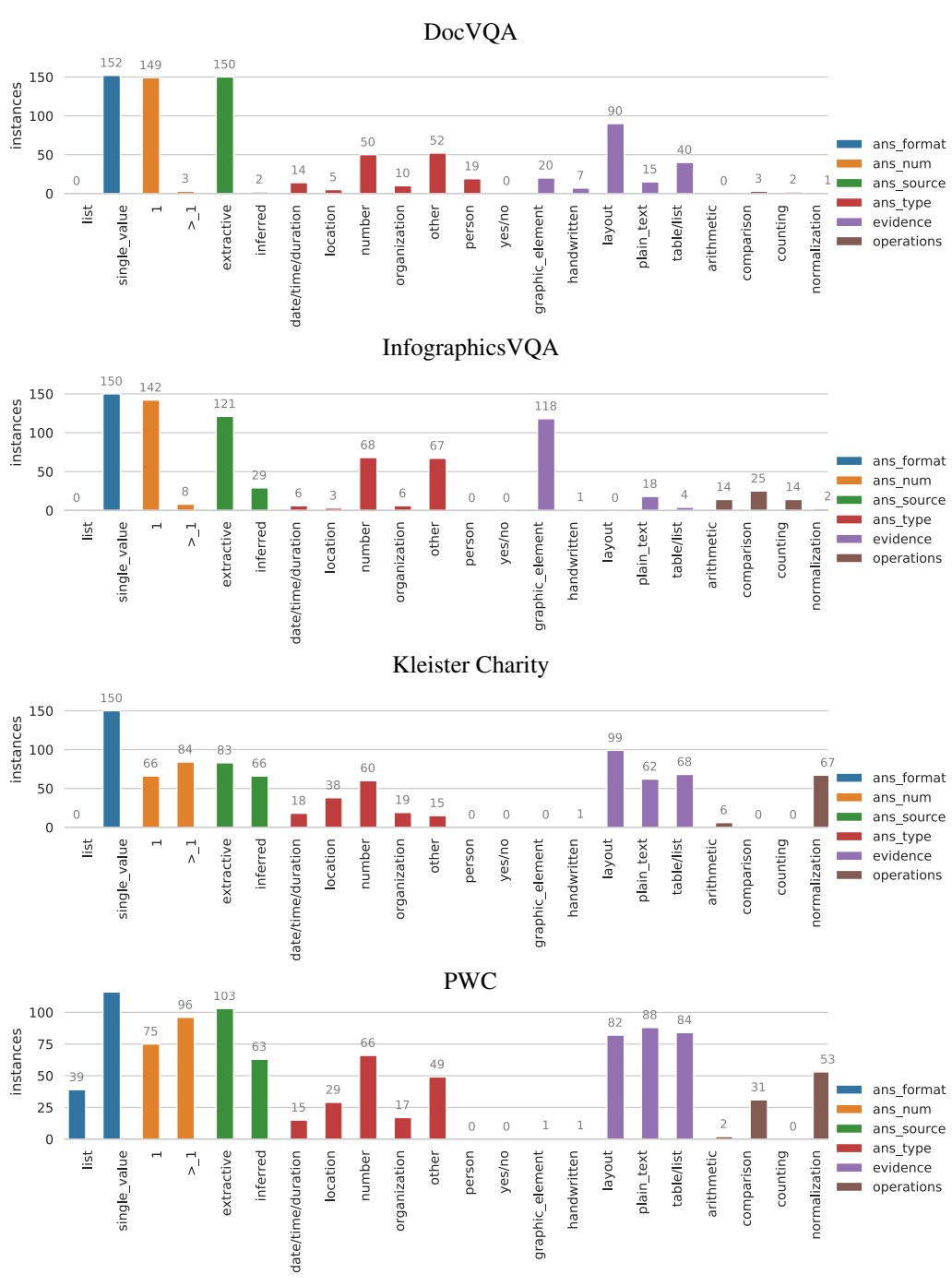

Figure 10: Number of annotated instances in each diagnostic subset category. DocVQA, InfographicsVQA, Kleister Charity, and PWC considered separately.

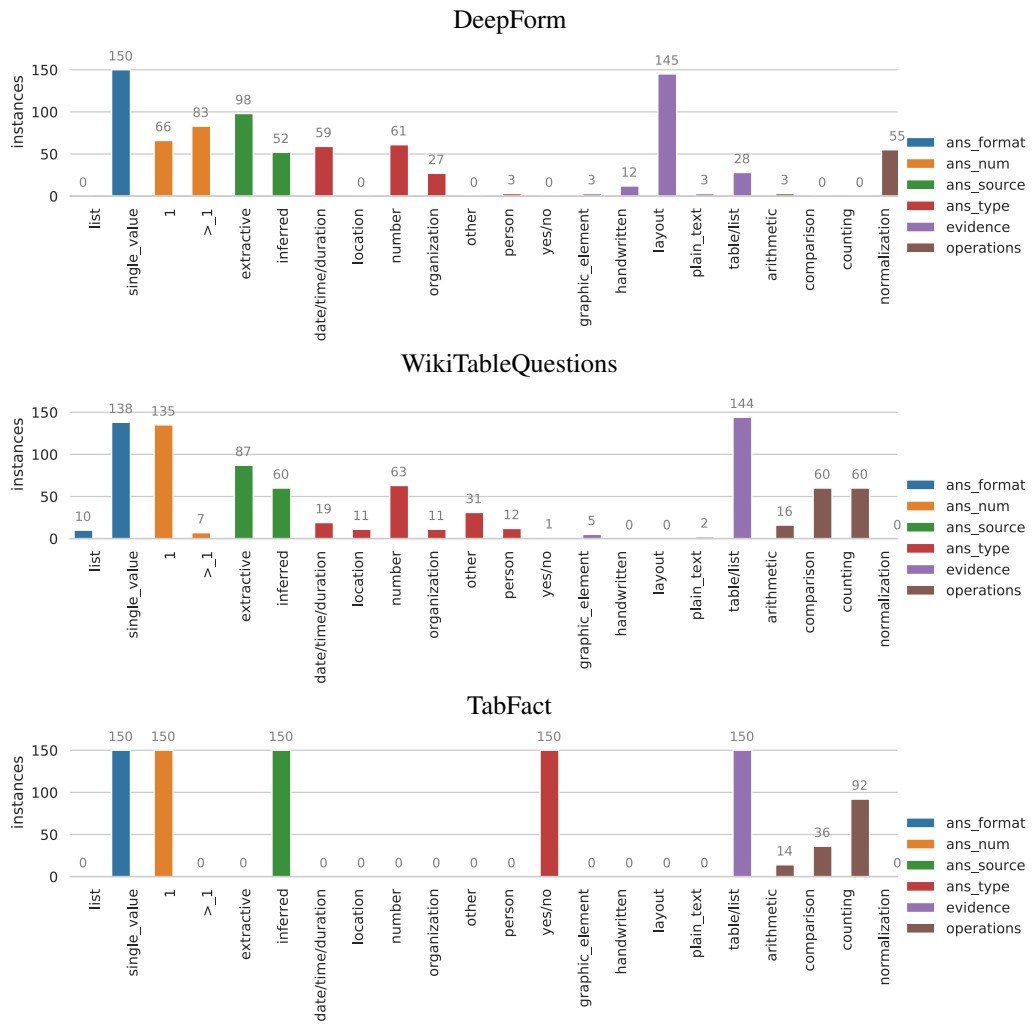

Figure 11: Number of annotated instances in each diagnostic subset category. DeepForm, WikiTable-Questions, and TabFact considered separatly.

- Every submission is required to have an accompanying description. It is recommended to include the link to the source code.

# I  Experiments — training details

The experiments were carried out in an environment with NVIDIA A100-40G cards, PyTorch version $1.8.1$, and the *transformers* library in version $4.2.2$.

The parameters were selected through empirical experiments with T5-Base model on DocVQA and InfographicsVQA collections. The T5-Large model was used as the basis for finetuning.

The training lasted up to $30$ epochs at batch $64$ in training, the default optimizer AdamW (lr = 2e-4), and warmup set to $100$ updates. Validation was performed five times per epoch, and when no improvement was seen for 20 validation steps (4 epochs), the training was stopped. The length of the input documents has been truncated to $6144$ tokens for all datasets (only Kleister Charity and PWC benefited from that change, for the rest of them $1024$ tokens is sufficient)[10] and the responses to $256$ tokens. Dropout was set to $0.15$, gradient clipping to $1.0$, and weight decay to 1e-5.

---

[10]The hard limit of 6k tokens results from the memory limitation of the used GPU.