# OpenReview forum: "DUE: End-to-End Document Understanding Benchmark"
_NeurIPS.cc/2021/Track/Datasets_and_Benchmarks/Round2 — NeurIPS 2021 Datasets and Benchmarks Track (Round 2)_

### Official Review · Reviewer_A26n · 2021-09-17
**Valuable work to process and normalize datasets.**

**Rating:** 6
**Confidence:** 4

**Strengths:**

* Processing all these datasets into a unified easy-to-use format and collecting all documents can be very valuable for any future research.
* The website and documentation seem intuitive and useful.
* The additional annotations can be very useful.


**Weaknesses:**

* The introduction discusses visual information and graphical elements which are included in some of the tasks but not in any of the evaluated baselines, and as the authors mention themselves in the results, it might in practice not be necessary. So it is unclear if this is part of what this benchmark is measuring or not.
* In general, the tasks are coming from different domains and and different qualities. The introduction reads as if doing better on this benchmark will result in models that perform better on any downstream document related task. But it is not clear to be that this correlation is true. It would be helpful to see some real tasks that succeeding on this benchmark correlates with doing better on them. Also in line 112: "it is crucial to consider these tasks together [...] may encounter in real=world applications" - I am not convinced by this paper that this is necessarily true specifically for these tasks.
* Following the previous point, it is unclear whether the spatial information is what needed in order to further improve performance on this benchmark or maybe it is just other NLP/vision/domain-specific capabilities. For example, maybe the T5+2D baseline already gets the most out of the document structure and the gap from the external best is due to other model improvements?
* Some descriptions to the baselines are missing (1) is T5+2D developed here or taken from a reference? How exactly is it implemented? (2) The OCR appears late in the paper without any explanation where in the process it is used (is it part of the preprocessing of documents?)
* The PWC task is very small (only 120 test examples) and the human performance is only 51. It feels a bit too noisy for evaluation. Also for WikiTableQ the human perf. is 77.
* No error bars. I am not convinced by the authors response to the checklist that it's only a rough estimate and "multiple runs are not necessary". Also, line 226 says "the training is run three times for each..". So I am confused. Which run was picked for results? Are the results in Table 3 for Dev or Test?
* How are the diagnostic subsets distribute across tasks? There's a risk of confusing domain performance etc. if some categories are dominated by specific tasks.

**Additional Feedback:**

-

**Clarity:**

* Overall the paper is pretty clear but it can use proofreading to correct a few typos.
* Table 1: align the dot to be in the same vertical position so, for example, .6, .12 and 1.7 will read correctly.
* figures 2 and 3 can use more human readable labels, and the alignment of the x-axis seems a bit off.

**Correctness:**

* Missing error bars and unclear if results in Table 3 are dev or test, and how was the reported model chosen (out of three runs?)

**Documentation:**

* Looks good to me

**Ethics:**

* Looks good to me

**Relation To Prior Work:**

* Seems good overall.
* Though, I believe that the KILT benchmark can be compared against as an example. In KILT the different tasks are performed over the same document (wiki articles). Therefore, the difference in perfomance between tasks cannot be attributed to different domains/ document quality/ etc. like in this paper.

**Summary And Contributions:**

This paper gathers 7 tasks in the area of document processing and understanding with the objective of creating a unified benchmark for evaluating model's ability to incorporate positional information in their predictions. The authors normalized the different tasks to have the same format and performed extensive data cleaning and added annotations for two of them. They also implemented baselines and measured human performance for datasets that were missing this information, and annotated certain features of examples to allow breakdown of the performance by categories.

---

> ### Author Response · Authors · 2021-09-27
> **Answers pt. 2**
>
> **Further improvements due to better spatial information usage**
>
> We agree with reviewer A26n that it is unclear what capabilities will be essential in the future improvements of the results on the datasets. Hence, we consequently devised diagnostic subsets to allow for such measurements. From Figure 3. it should be visible that for some types of reasoning, evidence, or operation, it is hard to achieve a good score with our T5+2D model. What is striking, even the results on layout-related data points are not very satisfying (~60% accuracy). The analysis suggests that there are plenty of different skills the model needs to acquire for performing well on the benchmark.
>
> **Descriptions of the baselines and OCR**
>
> Firstly, the OCR output is provided as part of a dataset (briefly described in #210 and Appendix H). However, it is also a part of the challenge to determine which OCR may work better, and we do not restrict researchers to employ only those popular OCR engines we used.
>
> Secondly, the T5+2D used 2D biases as proposed in TILT [1] and LayoutLMv2 [2]. Roughly speaking it is a simplified TILT model (with excluded UNet embeddings). The exact source code is based on HuggingFace Transformers and is submitted in supplementary materials and will be publicly released with the camera-ready version. The 2D positional biases are a generalization of 1D positional biases that the T5 model features and the specific implementation follows the work of [1].
>
>
> **The noisiness of the tasks**
>
> Firstly, while on the surface, the parameters (number of documents and human performance) look noisy for the task of PWC, it is crucial to notice that, in this reformulated scenario, one document contains on average 22 data points to be extracted. Hence, the total number of examples is much higher (2670 for the test set). Still, we acknowledge that this information is hard to deduce from Table 1 since the details of reformulations are covered in Appendix D, and the table's description does not suggest that the value refers to the number of documents. Therefore, we plan to add one column to Table 1, denoting the average number of data points in the dataset; furthermore, we will briefly describe how to interpret this in the caption.
>
> Secondly, the value of human performance should also be given a moment of thought, as the metric employed is not just pure F1 score but F1 based on grouped variables. This modification severely impacts the displayed value, as, e.g., models with per-datapoint accuracy of 85% can have on average a ~50% accuracy when groups of four are considered (the value may be correct but assigned to an incorrect group). In the latter case, every element in the group must be correct to qualify for one correct group. Therefore, we will use an appendix to also present results per each datapoint. Moreover, we are going to clarify the metric used soon.
>
> **Diagnostic subsets**
>
> Thank you for suggesting this improvement; it is a viewpoint of vital importance that we want to describe in the next revision using an additional page. The statistics will be presented in the diagnostic subsets section.
>
> **Error bars**
>
> The results for the tasks were presented on the test set, and an average of 3 runs was reported. The confidence intervals will be shown in the improved revision before the end of the rebuttal period.
>
>
> \
> [1] Powalski et al., Going Full-TILT Boogie on Document Understanding with Text-Image-Layout Transformer, ICDAR 2021.
>
> [2] Xu et al., LayoutLMv2: Multi-modal Pre-training for Visually-Rich Document Understanding, arXiv preprint, ACL 2021.

---

> > ### Comment · Reviewer_A26n · 2021-09-30
> > **Answer**
> >
> > Thank you for your detailed response

---

> ### Author Response · Authors · 2021-09-27
> **Answers pt. 1**
>
> We want to thank the reviewer for his valuable feedback and for scrutinizing the manuscript.
>
>
> **Importance of Vision**
>
> We acknowledge that the visual part of the benchmark may not be clearly explained, and to avoid further confusion, we want to point out several details related to how we will improve the manuscript in that matter.
> The importance of the visual aspects for datasets such as DocVQA was previously proven in ablation studies performed by [1], where authors concluded that their advantage comes from contextualized image embeddings. We are going to clarify this in an updated version of the paper. Moreover, we hope to include similar baselines with image embeddings before the rebuttal period ends as they are currently being trained, as suggested.
> As the 2d features help improve results on a graphical subset to some degree, the best scores achieved there are still at 26% level, suggesting that comprehending visual information is necessary to perform well. Consequently, the wording we use in line #301 `[...] part of the problems considered as visual can be in practice approximated by solely using the words’ spatial relationships` may be a little bit unfortunate, but hopefully is not suggesting the unimportant position of visual features in our benchmark.
> We will add the first point to the manuscript, and we will extend the result analysis section in the revision before the end of the discussion period to make the second point clear to the reader.
>
>
>
> **Real-world applications**
>
> Regarding this point, I believe there are two aspects that need to be addressed: one regarding the type of task and the other regarding domains and the potential problem of a domain shift when applied to real-world applications. Therefore, I will answer these two aspects separately.
>
> Firstly, by converting the tasks to the same QA-type task, while keeping their outputs differently structured (e.g., some tasks demand only boolean values, some tuples, some strings, and some have varied types), we ensure that the benchmark will be diverse enough to gather the attention of multiple research teams. However, if, for example, some prefer to work on just table-related QA, then submitting partial answers is welcomed, and the results will be displayed separately.
>
> Secondly, due to protecting business and personal information and further law enforcement (e.g., GDPR), the datasets published so far do not allow to align the domain easily to more real-world scenarios. It will probably remain like this in the future, leaving the research community with what is already published and allowed. It is thus still of persisting importance to gather diverse datasets and stimulate, as much as we can, the research community to overcome the challenges related to Document Understanding.
>
> We would be very pleased if, for example, the review pointed out a specific dataset we missed that may be much more realistic in that regard. Given the circumstances, we researched, gathered, and prepared the benchmark based on the available sources, with the desire to measure progress in the centralized QA setup, concentrating on issues related to the OCR, multi-modal understanding, and related to building unified architecture, but also, explicitly allowing for diversity in the source text's domain. Since there is a trade-off in constructing benchmarks, where one can choose to have either extremely domain-diverse datasets or single-domain ones, it is unclear from the literature which way is better under given constraints. In our situation, there is the option to have a cake and eat it too.
>
> We agree and are thankful to the reviewer for naming the domain aspect important, and we will also consider adding annotations related to the document's domain, as this may only further strengthen the idea of diagnostic subsets.

---

### Official Review · Reviewer_YsP1 · 2021-09-21
**End-to-end document understanding dataset**

**Rating:** 7
**Confidence:** 3
**Correctness:** The dataset construction is sound.
**Clarity:** The paper is clear and well-written.

**Strengths:**

The proposed benchmark joins several related tasks in an environment where models struggle (i.e. raw PDF reading), potentially stimulating research in a novel direction. The datasets are well selected, and accompanied by diagnostic tools that should encourage the development of models. There are large differences between baseline performance numbers on raw pdfs, baseline performance numbers on formatted text, and human performance; this is promising for the potential for this dataset to challenge models.

**Weaknesses:**

Potentially the largest weakness of the benchmark is that many or most examples only require reasoning over one form of data. Since the benchmark was stitched together from image datasets, table datasets, infographics datasets, and so on, few examples will benefit from reasoning that includes multiple modalities, e.g. tables *and* infographics. A smaller (related) weakness is the closed-domain nature of the benchmark, i.e. the fact that document retrieval is not required as the appropriate document to answer questions is given ahead of time. These are important problems for real-world document understanding scenarios, which models developed for this benchmark will not be tested on.

**Additional Feedback:**

N/A.

**Documentation:**

Yes.

**Ethics:**

No ethical concerns come to mind.

**Relation To Prior Work:**

The relation between this dataset and prior work is clear, as is the selection process for the datasets which the authors included in the benchmark. One minor omission is [1], which given the otherwise thorough overview in Table 5 should be included there.

[1] Gupta et al., 2020. INFOTABS: Inference on Tables as Semi-structured Data. ACL.

**Summary And Contributions:**

This paper introduces a new joint benchmark for end-to-end document understanding, combining several datasets and tasks. Distinct from prior work, this benchmark focuses on a very general setting where models only see a raw PDF version of the input data, along with bounding boxes etc. This significantly hinders common baselines -- for example, the performance of T5 on TabFact drops from near-SOTA to near-chance -- and as such provides a new and unresolved challenge. The authors accompany the dataset with several evaluative aids, including measurements of human performance on all datasets as well as manually annotated diagnostic datasets to identify problem cases for models.

---

> ### Author Response · Authors · 2021-09-29
> **Authors answer**
>
> First of all, thanks a lot for recognizing our paper and thoughtful analysis!
>
> **Multimodal reasoning**
>
> The datasets of DocVQA, Infographics, PWC, and Kleister-Charity consist of documents with tables, charts, and running text on the page. They all involve reasoning over multiple input sources, though the variability of the answer's origin is relatively high. For example, to answer questions in the Infographics dataset, it is expected to understand visual clues such as the color of the font, font size, arrows pointing from the text to the map, or identify symbols. Similarly, in the PWC, the results are primarily contained within the table; however, the full name of the model, metric used, or dataset is often provided in the caption or section related to methodology/results in the scientific paper. Finally, both DocVQA and the reports in Kleister-Charity may contain multiple questions that demand an understanding of the combination of running text, tables, forms, and handwritten signatures.
>
> To meet your request and clarify the importance of different modalities for each dataset, we will provide per-dataset diagnostic statistics in the next revision, to be released before the end of the rebuttal period.
>
> **Document retrieval**
>
> We agree with the reviewer that the tasks requiring retrieval are important in the real world. We believe, however, that this does not dismiss the importance of single-document understanding. There are multiple business cases where the document is given ahead of time and many separated situations where the document has to be retrieved from the collection. The issue we have with adding the retrieval task to the benchmark is that it demands measuring different properties that are often not aligned with our unified QA scenario. It seems like this particular problem lacks its own standard benchmark. This point of view is shared in the research community, and, e.g., tasks in the Document Visual QA Competition are split into "Single Document VQA" and "Document Collection VQA" where the latter challenges researchers with retrieval problems on top of document understanding ones [https://rrc.cvc.uab.es/?ch=17&com=tasks]. Both subtasks are essential for understanding documents; however, based on the analysis of results, one can admit that the shared task focused on single documents was met with higher participation, suggesting more research is carried out in this direction. On the other hand, it is also a more straightforward task that can help solve the more complex ones in the future.
>
> **Others**
>
> We will update our related works section with the mentioned paper.

---

### Official Review · Reviewer_4VHN · 2021-09-21
**Document understanding benchmark created using existing datasets**

**Rating:** 7
**Confidence:** 3
**Correctness:** Seems correct.

**Strengths:**

- Document understanding is a an important problem in multiple industries and this benchmark will clearly help in creating new solutions to this problem.
- Additional annotations and reformulations of existing dataset is a good way to large good quality datasets.
- Seems to have answered the questions raised by reviewers in round 1.

**Weaknesses:**

- No experiments were run with computer vision component, which would have shown how multimodal the dataset actually is.
- What is the size of each of document in the dataset, as in how many words and when T5 truncates the input to 1024 tokens, how often does it remove the answer?
- The output of OCR isn't clear for the cases of Tables.
- The datasets used are all of single modality systems, which means only one modality is needed to get the answer to the question. Multimodal systems are more like TAT-QA, HybridQA and OTT-QA where tables and text in the document needs to be processed to find the answer.

**Additional Feedback:**

NA

**Clarity:**

- Decent writing, but could be little more clearer on the algorithmic aspect.

**Documentation:**

We documented.

**Ethics:**

It is collating lot of existing datasets and reformulating and cleaning them. So no ethical concerns.

**Relation To Prior Work:**

- I think FinTabNet[1] work could be cited in the paper:

[1] Zheng, Xinyi, et al. "Global table extractor (gte): A framework for joint table identification and cell structure recognition using visual context." Proceedings of the IEEE/CVF Winter Conference on Applications of Computer Vision. 2021.

**Summary And Contributions:**

The paper introduces a new end-to-end document understanding benchmark created using existing datasets by putting them all in the same format. The authors reformulated existing datasets on TableQA and Table verification and PWC to make it a document understanding problem. A lot of cleaning was done on another four more datasets before adding them to the benchmark. Authors showed how some of existing deep learning methods perform on this new benchmark.

---

> ### Author Response · Authors · 2021-09-27
> **Answers from authors**
>
> We want to thank the reviewer for his valuable feedback and for scrutinizing the manuscript.
>
>
> **Multimodality**
>
> In the beginning, I would like to point out that the multiple modalities of the datasets are text, layout, and vision. Another important argument is that one can support the claim regarding the dataset's multimodality in different ways, not necessarily by implementing visual components, which I will describe below.
>
> While our experiments cover two modalities (text and layout, i.e., text spatial arrangement), the previous work described in the literature already proved the importance of the visual component for datasets such as DocVQA [1]. Hence, we retreated from the costly reimplementation of such changes to focus on providing higher-quality datasets and diagnostic subsets. Further analysis of the results on the diagnostic subsets allows one to formulate a definite statement that models without visual components achieve poor results (~26%) on examples where evidence is presented in the graphic form.
>
> So, the datasets we gathered are multimodal, and to perform well on them, there is a need to comprehend visual modality, which was proven to some degree before, and is an obvious conclusion from our Figure 3.
>
> Additionally, to meet your request and clarify how much visual aspects contribute, we will provide per-dataset diagnostic statistics in the next revision, to be released before the end of the rebuttal period
>
> **Size of the documents**
>
> We admit that the average size of the document may be important information to provide in Table 1. Moreover, to answer that question, we analyzed the system's results depending on the document length for the Kleister-Charity dataset, as it features the longest documents. We will add these results to the extended version of the manuscript that we will submit before the rebuttal period's end.
>
> **OCR for Tables**
>
> The OCR information we provide for each dataset consists of two pieces of information: text and its bounding box as read by the OCR engine. In the cases of tables, the OCR reads every word and provides its bounding box. The bounding boxes of multiple words in the same cells are not merged nor post-processed, but it is allowed for participants to do it and use this information in any way they like. Furthermore, we leave the freedom of choosing the best OCR to the participants, as the PDF documents are provided for each example.
>
> **Other datasets**
>
> The datasets of DocVQA, Infographics, PWC, and Kleister-Charity consist of documents with tables, charts, and running text on the page. They all involve reasoning over multiple input sources, though the variability of the answer’s origin is relatively high. As multiple modalities can appear in the same document, we did not consider datasets where the modalities are provided separately as distinct input sources. Considering multihop QA datasets may be an exciting direction for further expansion of the benchmark. Still, since the unification of multiple datasets into a common format is the dominant theme of DUE, it is unclear how we can unify multihop QA datasets with others. We are thankful to the reviewer for inspiring us to consider these ideas.
>
> We considered TAT-QA and HybridQA in Table 5. However, we rejected them based on either the lack of availability of the input sources or the formulation incompatible with our unified end-to-end approach. The OTT-QA demands retrieval, and hence may be an excellent source for another benchmark to measure progress in the field of document collection understanding. The FinTabNet dataset is slightly related to the Document Layout Analysis, and we will cite it in the associated section. Please note that Appendix B contains an in-depth description of the dataset selection process.
>
> We hope that announced changes, to be introduced in the next few days, and provided comments will satisfy the objections that were raised. If it doesn’t suffice, we would be helpful for other suggestions during the rebuttal period.
>
> [1] Powalski et al., Going Full-TILT Boogie on Document Understanding with Text-Image-Layout Transformer, ICDAR 2021.

---

> > ### Comment · Reviewer_4VHN · 2021-10-02
> > **Increased score after going through the rebuttal**
> >
> > The rebuttal is convincing.

---

### Official Review · Reviewer_HWA2 · 2021-09-24
**Interesting benchmark**

**Rating:** 7
**Confidence:** 4

**Strengths:**

I really liked the paper. The problem with having a limited space to convey a lot of information automatically leads to issues down the line, where somebody says : hey, more details on the baselines, and somebody else: not enough descriptions on the datasets. I personally would dedicate most space to the datasets themselves and less on the baseline that needs to be broken anyway by future participants.

Biggest strength: - The annex answers most foggy areas that couldn't be addressed in the paper that popped up when reading it.

- There are many datasets unified
- Format is pretty clear (thanks annex)
- Baseline seems reasonable
- There was a non-trivial amount of manual work poured in the datasets


**Weaknesses:**

- Like others before me, I wonder how useful in reality is a benchmark that packages to a high-level many not-quite-that-similar datasets. For example, I had a problem at work where I needed to understand a somewhat stable type of documents. I looked for tasks that are similar, and systems that solved this particularly close tasks. Looking back, I don't think I would have checked a unified benchmark (like this one) unless it really pointed out that particular set of tasks (or single task) that was close to my work task. Furthermore, I most likely wouldn't develop a system that worked on all tasks given that I was focused in particular on, say, one of them. Just like in the case of SUPERGLUE, where I'd just check, on average, which type of transformer works best, and not a particular system, as there are many variable tasks in there. But maybe that's particular to me.

- The website, while stylish, does not allow me to sort the best model by the WikiTable Questions dataset, for example. It is likely that in my work I would like to see the best performing systems sorted by one of the tasks that I find most similar to my current problem. I can't do that here, I have to manually check each system in turn. I know it "looks good" but a benchmark should focus on usability (e.g. like the results reported on paperswithcode.com)

- The website again, I can't see the date when a system was added. I would like to know that information (like, is it old, is it new?)

- The website, scores in the table are not sortable..

- https://duebenchmark.com/analysis gives me a nice 404

- I will not pick at some details of the paper. Overall this revision looks clear and good enough to read to me, there's a sufficient amount of info and a lot in the annex.

**Additional Feedback:**

N/A

**Clarity:**

Well written. I would have structured it differently, with more focus on the datasets and less on the baselines, but there's always a mix that will not be to everybody's liking.

Regarding the site (not the paper):
What I would improve on is the process of you (the authors) getting submissions on the benchmark. At the moment, for me, it seems I'd have to do quite some research to understand everything I'd need to do, from the data offered that's not explained at all on the website ( https://duebenchmark.com/data ), to the submission process which is not detailed - there are just some rules ( https://duebenchmark.com/rules ). In this case I would not take the time to make an account to see if there's anything more after the log-in screen.


**Correctness:**

I think the paper (benchmarks) checks all the tick-boxes to a sufficient degree. I have no major issues here.

**Documentation:**

Could be improved, see clarity section.

**Relation To Prior Work:**

Minimal but given the space constraints, I think it's okay.

**Summary And Contributions:**

The authors propose an end2end benchmark for document understanding, providing the website, process, details, datasets, and including a baseline for this task.

---

> ### Author Response · Authors · 2021-09-29
> **Authors answer**
>
> Thank you so much! We appreciate your acknowledgment. Also, thank you for being reasonable in evaluating the manuscript's composition and articulating your demands.
>
>
> **Real-life vs. conglomerate benchmarks**
>
> It is an interesting point of view. We believe that it really touches the subject of where are the limits of beneficial similarity between different tasks considered in the same conglomerate benchmark. Therefore, I will elaborate on that to present our stance.
>
> It is true; it is not apparent for this benchmark and many others how it will drive improvements on other real-world tasks, hence our motivation to unify them into end-to-end tasks and provide support for a precise measurement where each task helps.
>
> Bearing in mind that our studies have shown that, e.g., unsupervised pretraining of 2D features improved results on multiple of them, this suggests there is something in these tasks that makes them similar and allows for transferring knowledge related to the document understanding to others. As a result, when we look from the scientific perspective, we see that, since there is a notion of layout more prominent than each dataset itself, it is worth investigating where we can settle in this space of joined interests.
>
> Finally, it is also of great significance that future researchers working with DUE can easily convert their tasks into the same "QA over PDF" format we proposed and leverage 100% of the pretrained systems knowledge. Compared to SUPERGLUE, this is a notable improvement that can make a difference in approaching downstream tasks.
>
> **Website**
>
> While the website is still a work in progress planned to be published soon, we admire and accept your feedback. To show that the website is constantly improving, we released some minor updates to answer the mentioned issues. However, more work is scheduled after the rebuttal period.
>
> **Dataset vs. baseline balance**
>
> Since the area chair's decision in round 1 was focused on a lack of in-depth analysis of the baseline's performance on the benchmark, we committed ourselves to improve this aspect; however, we also believe it should never be regarded as the main contribution of a benchmark paper. On the other hand, please note that we will be using an additional 1 page to describe datasets better, add more information to table 1, and present statistics about the diagnostic subsets. This will hopefully help us dedicate adequate space to the datasets themselves.

---

### Author Response · Authors · 2021-09-30
**Updates The updated version of the manuscript**

Thank you once again for the valuable discussion, constructive remarks, and advice. As noticed by HWA2, it is hard to balance between different expectations of the reviewers. Nevertheless, we did our best to consider all the remarks from the present and previous round of reviews. We are extremely helpful and believe the paper benefited a lot from your suggestions.

We managed to introduce numerous of the requested changes already during the discussion period. Changes, highlighted with red in the updated version of the manuscript, include, among others:
- analysis of the diagnostic categories distribution across particular datasets (Section 3.2 and Figures 10-11 in the Appendix); as mentioned, the general framework we designed can be used not only to analyze model performance but also to characterize the datasets themselves
- clarification of the preprocessing with optional OCR step (Section 4)
- more precise information on used 2D bias architecture (Section 4)
- correction of experiment details with respect to the number of runs with different seeds (part of the paper was not up-to-date); additionally, Table 3 now covers information on mean and standard deviation across different runs
- clarification of why we believe multimodal aspect is vital when attempting the datasets covered by benchmark (Section 4.3)
- requested consideration of performance w.r.t. the consumed input sequence length on the example of Kleister Charity (Section 4.3) and clarification regarding used input length in our encoder-decoder baseline

Additionally, several changes regarding the website functionality are going to be released later this day. We hope these and other ongoing changes, as well as provided comments, satisfy the objections raised and kindly ask the reviewers to reconsider their evaluation.

---

### Decision · Program_Chairs · 2021-10-09

**Decision:**

Accept

**Comment:**

The work presents a new end-to-end document understanding benchmark, which all reviewers think is interesting. Additionally, the authors have set up a leaderboard website with some useful features. Despite some concerns from reviewers, the authors make a clear and convincing rebuttal.